# Role of Nodulation-Enhancing Rhizobacteria in the Promotion of *Medicago sativa* Development in Nutrient-Poor Soils

**DOI:** 10.3390/plants11091164

**Published:** 2022-04-26

**Authors:** Noris J. Flores-Duarte, Enrique Mateos-Naranjo, Susana Redondo-Gómez, Eloísa Pajuelo, Ignacio D. Rodriguez-Llorente, Salvadora Navarro-Torre

**Affiliations:** 1Departamento de Microbiología y Parasitología, Facultad de Farmacia, Universidad de Sevilla, García González 2, 41012 Sevilla, Spain; nflores@us.es (N.J.F.-D.); epajuelo@us.es (E.P.); irodri@us.es (I.D.R.-L.); 2Departamento de Biología Vegetal y Ecología, Facultad de Biología, Universidad de Sevilla, Apartado 1095, 41012 Sevilla, Spain; emana@us.es (E.M.-N.); susana@us.es (S.R.-G.)

**Keywords:** legumes, plant growth-promoting rhizobacteria (PGPR), nodulation-enhancing rhizobacteria (NER), biofertilizers, nutrient poverty, *Medicago*, abiotic stress, degraded soils

## Abstract

Legumes are usually used as cover crops to improve soil quality due to the biological nitrogen fixation that occurs due to the interaction of legumes and rhizobia. This symbiosis can be used to recover degraded soils using legumes as pioneer plants. In this work, we screened for bacteria that improve the legume–rhizobia interaction in nutrient-poor soils. Fourteen phosphate solubilizer-strains were isolated, showing at least three out of the five tested plant growth promoting properties. Furthermore, cellulase, protease, pectinase, and chitinase activities were detected in three of the isolated strains. *Pseudomonas* sp. L1, *Chryseobacterium soli* L2, and *Priestia megaterium* L3 were selected to inoculate seeds and plants of *Medicago sativa* using a nutrient-poor soil as substrate under greenhouse conditions. The effects of the three bacteria individually and in *consortium* showed more vigorous plants with increased numbers of nodules and a higher nitrogen content than non-inoculated plants. Moreover, bacterial inoculation increased plants’ antioxidant activities and improved their development in nutrient-poor soils, suggesting an important role in the stress mechanisms of plants. In conclusion, the selected strains are nodulation-enhancing rhizobacteria that improve leguminous plants growth and nodulation in nutrient-poor soils and could be used by sustainable agriculture to promote plants’ development in degraded soils.

## 1. Introduction

Legumes are a family of plants (*Fabaceae/Leguminosae*) formed by 765 genera and around 19,500 species [1]. This family of plants is characterized by the symbiotic relationship with rhizobia, a group of α- and β-proteobacteria including several genera, among others, such as *Bradyrhizobium*, *Ensifer*, *Mesorhizobium*, *Rhizobium*, and *Sinorhizobium* [2]. In this symbiosis, plants offer both a niche and carbon source to the bacteria while the latter provides NH_4_^+^ by its ability to reduce atmospheric N_2_ within the nodules [3]. Thanks to this association, legumes are pioneer plants that can grow and colonize degraded environments, overcoming abiotic stresses (nutrient-poor soils, saline soils, polluted soils, etc.) [4,5] and contributing to enrich degraded soils, with nitrogen improving their quality and fertility [6,7]. For this reason, legumes are used as a transition plant in intercropping to recover soil quality after a crop cultivation [8,9]. Nevertheless, nodulation and the nitrogen fixation effectiveness can be affected by abiotic stress [10,11], which also conditions the rhizobial population present in these kinds of soils [11].

In the rhizosphere, a higher concentration of bacteria exists around roots due to the exudates of plants which bacteria use as a source of nutrients [12,13,14]. Several of these rhizospheric bacteria possess plant growth-promoting (PGP) properties that help plant development and are known as plant-growth-promoting rhizobacteria (PGPR). PGPR assist plants by means of direct and indirect mechanisms such as nutrient (phosphorous, iron, and nitrogen) acquisition, phytohormones production, ethylene level modulation, and the production of lytic enzymes against phytopathogens [5,15,16]. Among these, the solubilization of phosphorous, siderophores production and nitrogen fixation are the major beneficial properties involved in nutrients acquisition. Thus, the presence of these properties in the bacteria helps plants to solubilize insoluble phosphates, improve iron uptake with siderophores, and increases the nitrogen content of the plant through nitrogen fixation [14]. In addition, PGPR can synthesize and produce phytohormones such as auxins (mainly indole-3-acetic acid, IAA), stimulating root elongation and improving nutrient acquisition [14]. The stress level in plants can also be modulated by PGPR’s amino-cyclopropane carboxylic acid (ACC) deaminase activity, an ethylene’s precursor degrading enzyme [14,17].

Among all PGPR, there is a non-rhizobia group which, in addition to its plant promoting properties, improves the symbiotic relationship between legumes and rhizobia enhancing nodulation, known as nodulation-enhancing rhizobacteria (NER) [11]. This kind of bacteria helps the plant to form nodules thanks to PGP properties—IAA production and ACC deaminase activity, the main traits implicated in this nodulation improvement [11].

The aim of this study is to enhance the growth of legumes in soils with nutrient poverty by means of the NER activities in order to recover degraded soils and improve their quality and fertility. For that, we selected soils of the Rio Piedras estuary (Huelva, Spain) as nutrient-poor soil, *M. sativa* as a model legume, and *Ensifer medicae* MA11 as the rhizobium to carry out the experiments. The interaction *Medicago*-*Ensifer* is well-known [18], and the strain MA11 was isolated from a degraded environment, showing nodulation capabilities under these kinds of conditions [19].

To accomplish the main aim, the following objectives were pursued: (i) isolation of rhizobacteria from the rhizosphere of *Medicago* spp. from the Rio Piedras estuary; (ii) first screening based on the ability to solubilize phosphate; (iii) identification and characterization of the phosphate solubilizing bacteria displaying additional PGP properties and enzymatic activities; (iv) selection of the best isolated PGPR; (v) determination *in vitro* of the role of the selected PGPR in the germination and nodulation in *M. sativa*; (vi) determination of the effect of selected PGPR in the growth and the physiological status of plants of *M. sativa* grown in soil from the Rio Piedras estuary under greenhouse conditions.

## 2. Results

### 2.1. Isolation and Characterization of Rhizosphere Bacteria

Cultivable bacteria were isolated from rhizosphere soil of *Medicago* spp. plants from the Rio Piedras estuary. This soil was composed of 92% sand, a very low quantity of organic material (0.98%), and low values of nutrients characterized as a nutrient-poor substrate (Table 1). In addition, pH was alkaline (8.5), and the soil showed a high electric conductivity, corresponding to estuarine sediments.

A total of 71 different bacteria were isolated based on the colonies’ morphology and colour, cell morphology, and Gram staining. Due to the low quantity of P in soils (Table 1), phosphate solubilizing bacteria were searched as a first screening and 14 bacteria were selected (Appendix A). Among them, 86% were Gram-negative rods, 7% Gram-positive rods, and 7% sporulated Gram-positive rods (Appendix A).

To identify the different isolates, the strains with different colony aspects and different profiles in the Box-PCR (data not shown) were identified by 16S rRNA gene sequencing, and genus *Pseudomonas* was the most representative (Table 2). The rest of the isolates belonged to genus *Chryseobacterium*, *Priestia*, *Bacillus*, and *Butiauxiella*.

In relation to the bacterial characterization, PGP properties were checked in the rhizobacteria and all of them showed at least three properties (Figure 1A; Appendix A). In addition to phosphate solubilization, all of the isolates were able to produce IAA (Figure 1A). Strains L15, L4, and L8 were the best phosphate solubilizers (18, 14, and 14 mm, respectively), while L15, L10, and L2 produced the highest amount of IAA (11.133, 6.759, and 6.522 mg·L^−1^, respectively) (Appendix A). Eighty-six per cent of the rhizobacteria could fix nitrogen; in addition, siderophores producers had the strain L4 (26 mm of halo; Appendix A) as the best one. ACC deaminase activity was present in 50% of the strains, and the highest activity was found in strain L8 with 2.087 μmoles α-ketobutyrate·mg protein^−1^·h^−1^, followed by L14 with 1.684 μmoles α-ketobutyrate·mg protein^−1^·h^−1^ (Appendix A). Finally, biofilm formation was exhibited by L1 and L11, representing 14% of the total bacteria (Appendix A).

Enzymatic activities were also studied to characterize the isolates (Appendix A). Enzymes such as cellulase, pectinase, protease, and chitinase were detected in the different bacteria. The most abundant enzymatic activity was protease, present in 21% of the isolates, followed by pectinase activity, present in 14% (Figure 1B). On the other hand, cellulase and chitinase activities were found in 7% of the bacteria (Figure 1B). Despite only 21% of isolated bacteria presenting these enzymatic activities, they showed more than one activity, L3 being the strain with more positive enzymatic activities (Appendix A).

Based on these results, strains L1, L2, and L3 were selected as inoculants for *in planta* assays since they had high values in PGP properties and were the only ones that showed the tested enzymatic activities (Appendix A).

### 2.2. Effects of Selected PGPR in the Germination, Growth, and Nodulation of M. sativa Plants In Vitro

The selected strains were inoculated to *M. sativa* seeds to observe their effect on germination. Figure 2A shows that inoculated seeds had a higher percentage of germination than non-inoculated control seeds, particularly those inoculated with the bacterial *consortium* (CSL treatment), reaching 100% of germinated seeds *versus* 81% of non-inoculated ones. When tested individually, the three PGPR had a similar germination percentage improvement compared with control seeds, although not as high as the *consortium*.

In addition to germination, as a preliminary test before greenhouse experiments, the effect of selected PGPR´s inoculation on plant growth and nodulation was studied *in vitro*. Inoculated plants had more biomass, both shoots and roots, than control plants without inoculation (Figure 2B). Individually, strain L2 inoculation correlated with *M. sativa* highest dry weight values, particularly in shoots, showing a biomass increase regarding the control of 240% and regarding MA11 inoculation of 54%. However, the more vigorous plants were those inoculated with the *consortium*, showing an increase on dry weight regarding the controls of 280%. Regarding nodulation, the selected PGPR inoculated individually increased the number of nodules per plant, although the *consortium* once again showed the highest number, increasing two-fold the nodulation by MA11 (Figure 2C).

### 2.3. Effects of Selected PGPR in the Physiological Status and Nodulation of Plants under Nutrient-Poor Soil Stress

*M. sativa* physiological and nodulation changes induced by *in vitro* inoculation with selected PGPR were also tested under greenhouse conditions using as substrate a nutrient-poor soil collected from the Rio Piedras estuary. Both the length and size of plants and number of leaves were significantly increased with PGPR’s individual inoculation. The highest values were obtained with the *consortium* inoculation (Figure 3A,B; Appendix A). Similarly, the control plants’ total biomass was surpassed by the inoculated plants’ biomass, particularly those inoculated with the *consortium* (Figure 3C). PGPR also improved plant nodulation in nutrient-poor soil. Plants inoculated with the *consortium* showed a significantly higher number of nodules than those inoculated individually (Figure 3E). Among plants with single inoculation, strain L2 showed the highest values regarding the tested physiological parameters, although no statistically significant difference was observed (Figure 3A–C,E). N content in plants also increased in inoculated plants, particularly with the *consortium* of PGPR (Figure 3D).

Several parameters were determined in order to establish the physiological state of the plants, namely, the net photosynthetic rate (A_N_), the stomatal conductance (g_s_), the intercellular concentration of CO_2_ (C_i_), the efficiency of photosystem II (Fv/Fm), and total chlorophyll content. Regarding gas exchange in plants inoculated with different strains, the net photosynthetic rate was higher in those inoculated with L2 strain and the *consortium* with similar values (Figure 4A). In the case of the stomatal conductance, plants inoculated with L1 strain showed the highest value, followed by plants inoculated with the *consortium* without significant differences (Figure 4B). The values of intercellular CO_2_ concentration in plants were very similar in all of the inoculation conditions, plants inoculated with strain L1 outlined significantly (Appendix A). In addition, results related to plants’ photochemistry were also statistically very similar among the different inoculation treatments, where the performance of plants inoculated with CSL stood out over the others, showing the highest values in all of the tested parameters (Figure 4C; Appendix A). On the other hand, the total chlorophyll content results showed significant differences among the different inoculations (Figure 4D). Individually, plants inoculated with strains L2 and L3 produced amounts of chlorophyll with significant difference compared with controls; however, the highest increase appeared in plants inoculated with the *consortium*, with five-fold more chlorophyll than non-inoculated plants and three-fold more than plants inoculated with MA11.

Finally, in order to see the effect of selected bacteria on the stress mechanisms of plants due to the low quantity of nutrients in the substrate, the activities of antioxidant enzymes were measured in leaves. In general, the activities of all tested enzymes increased with bacterial inoculation compared with non-inoculated plants, except for catalase activity (Figure 5A–D). This enzyme activity was statistically similar in all of the treatments apart from CSL inoculation, whose plants showed significantly more activity (Figure 5A). Inoculation with the *consortium* increased the activities of all of the enzymes, although the inoculation with strain L2 also significantly increased the activity of ascorbate peroxidase and guaiacol peroxidase (Figure 5B,C). Superoxide dismutase was the enzyme with highest activity increase in all of the treatments, while catalase showed the lowest activity change.

## 3. Discussion

For decades, natural and human activities have caused negative effects in lands, giving rise to degraded soils characterized by poor quality, a decrease in fertility and nutrients content, and an increase of abiotic stresses such as salinity, heavy metals content, or drought. The presence of abiotic stresses in soils reduces plant biodiversity and creates an imbalance of nutrients [20]. Nutrient-poor soils produce a loss of microbial biodiversity and fertility and prevent the growth and development of plants, decreasing the ecological value of estuaries [21,22]. In addition, low fertility is one of the main limitations in crop production [23], so the loss of fertile soils is a worldwide challenge regarding food security [24].

A possible solution for the ecological restoration of agricultural lands is the use of legumes as cover crops. Legumes are beneficial in agricultural practices because their use reduces greenhouse effect gases, reduces the need for nitrogen based artificial fertilizers, avoids soil erosion, and, most importantly, provides organic compounds and nitrogen, increasing the fertility, nutrient content, and quality of soils [25,26,27].

The main goal of this work is the nodulation and development improvement of legumes, particularly *M. sativa*, in environments with nutrients poverty in order to use them as pioneer crops to recover the quality of degraded soils, promoting a sustainable agriculture. *M. sativa* was selected, in addition to the benefits that legumes provide to soils, because it is an important forage crop digestible for animals with great yield and high nutrient values [28]. For nodulation to be effective under stress conditions, proper rhizobial strains with high resistance towards stress must be used. In our case, *Ensifer medicae* MA11 had previously been shown to effectively nodulate *Medicago* plants under arsenic stress [19]. In addition to using a tolerant rhizobial strain, the isolation, characterization, and selection of PGPR and NER from the rhizosphere of *Medicago* spp. plants growing in a nutrient-poor estuarine soil were performed, since PGPR can assist plant growth through direct and indirect mechanisms [14,15,16], and NER can improve legumes nodulation [11].

Rhizospheric soil collected from the Rio Piedras estuary was mainly composed of sand, making it a light and drain soil. Furthermore, this soil had less than 1% of organic material and a low quantity of nutrients. These characteristics correspond to the definition of nutrient-poor soil [29,30,31], so the estuary of Rio Piedras could be considered as a degraded environment. The knowledge of the microbial population of this degraded soil is very important to find PGPR and NER able to promote legumes growth under the scarcity of nutrients [32]. In this work, 71 cultivable bacteria were isolated and, due to the low level of phosphorus in the soil, a first screening was performed, selecting those bacteria capable of solubilizing phosphate, reducing the selection of isolates to 14. Phosphorus is one of the most limiting factors for plant development, and phosphate solubilizing bacteria make it available to plants through the secretion of phytases, phosphatases, and organic acids into the soil [33]. The diversity of root-associated bacteria depends on the environment and its stress level [34,35,36,37,38], showing a lower number of microorganism taxa and fewer interactions among taxa in degraded soils with erosion [35,39]. This low diversity was also observed in rhizosphere of *Medicago* spp. From the Rio Piedras estuary at the genus level, *Pseudomonas* being the most representative (Table 2). The presence of *Pseudomonas* as the most represented genus among the cultivable bacteria in this study is not a surprise since it is ubiquitous in soil and has well known genetic, environmental, and physiological adaptability to survive in any environment [40]. In addition, rhizosphere bacteria belonging to genus *Pseudomonas* had been isolated from other legumes such as peanut, soybean, and broad bean [41,42,43,44]. The rest of the genera isolated in this work, namely, *Chryseobacterium*, *Priestia*, *Bacillus*, and *Buttiauxella*, have been also described as associated to legumes by other authors [45,46,47,48,49,50].

In addition to phosphate solubilization, isolated bacteria showed individually several PGP properties (Figure 1A). IAA is an auxin, a phytohormone involved in numerous processes in plant development, mainly in root elongation facilitating nutrient absorption by plants [51]. Moreover, IAA is also involved in nodule formation because it intervenes in the relationship between rhizobia and legume, and the functionality of the nodule gets to delay the senescence by the interaction with the bacteroid inside the nodules [11]. This property was observed in all isolates, making all of them good candidates to improve the nodulation of legumes growing in degraded estuaries. Siderophores production is another important PGP property in degraded soil with nutrient deficiency because they have affinity for iron, forming a complex that can be assimilated by plants [52,53]. Furthermore, the production of siderophores is also related to biotic control due to competition for Fe with phytopathogens [12]. A total of 87% of isolated strains in this study were grown in minimal medium without a nitrogen source, indicating that they could fix nitrogen, which would additionally increase the amount of nitrogen fixed within the nodules [54]. ACC deaminase, detected in half of the rhizobacteria (Figure 1A), allows bacteria to modulate the ethylene concentration in plants since this enzyme hydrolyzes the ethylene’s precursor [55,56]. With this modulation, bacteria promote plant growth under stress conditions [57] and even improve the nodulation and the functionality of the nodules because ethylene is also involved in nodule senescence [58,59,60]. Finally, although biofilm formation was the less represented property among isolates, it could help plants to grow in degraded environments since biofilm concentrates bacteria coating roots and facilitates important processes like nitrogen fixation, the mineralization of organic N and P, and nutrients absorption [61].

The secretion of lytic enzymes such as chitinases, lipases, pectinases, or proteases acting against the wall of phytopathogens are also interesting traits to enhance plant development [55]. In this respect, only L1, L2, and L3 showed enzymatic activities for protease, pectinase, and chitinase. Moreover, strain L3 also showed cellulase activity, important to degrading the vegetal cell wall and facilitating the rhizobia entry in roots [62,63]. These three strains, *Pseudomonas* sp. L1, *Chryseobacterium soli* L2, and *Priestia megaterium* L3, which also showed five of the six tested PGP properties with high values individually and all of the PGP properties altogether, were selected to continue with plant experiments.

The inoculation of *M. sativa* seeds with selected strains showed that all of them increased the number of germinated seeds individually, although inoculation with the three (*consortium*) showed the highest number of germinated seeds. This positive effect in *M. sativa* seed germination was also reported in [64], where *Bacillus* spp. strains improved the number of germinated seeds.

The results concerning plant development and nodulation were similar both *in vitro* and under greenhouse and nutrient-poor conditions. Inoculated plants showed an increase in both shoots and roots development, the increase in those inoculated with the *consortium* being the highest one. Supporting these results, several authors verified that the inoculation of *M. sativa* plants with PGPR improved plant biomass and length in nutrient-poor substrate, both *in vitro* [65] and in greenhouse conditions [66] and even in the presence of other abiotic stresses such as salinity and heavy metals [28,64,67,68]. In addition to plant development, the nodulation in plants was also higher in inoculated groups, in agreement with the results obtained in other studies where the nodulation in *M. sativa* was tested under stress conditions [69]. The fact that co-inoculated plants had a greater number of nodules could be related to the nitrogen content in plants since they also showed higher values of nitrogen [28].

The positive effect in plant physiology was also observed in photosynthetic parameters since inoculation with the *consortium* improved the photosynthetic status in plants under nutrient poverty stress. Similarly, in *M. sativa* plants growing in a substrate with a low quantity of phosphorous, the inoculation with *Priestia megaterium,* formerly *Bacillus megaterium* [70], increased the chlorophyll content [66], and in *M. sativa* plants under heavy metals stress, inoculation was involved in higher values of physiological parameters such as the ETR, Fv/Fm, Φ_PSII_, A_N_, and g_s_ [66].

Finally, the stress level in plants growing in nutrient-poor estuarine soils was determined by the measurement of antioxidant enzymatic activities. The inoculation with the *consortium* of the selected bacteria showed a significant increase in these enzymatic activities. This increase was also observed in other studies with *M. sativa* plants under different abiotic stresses [28,64], suggesting that bacterial inoculation could elevate the stress response in plants, improving and recovering the plant status in degraded environments.

According to the results, the inoculation of *M. sativa* with the *consortium* of the three selected strains showed more vigorous plants with more nodules than the single inoculations, suggesting that the *consortium* had the higher effect on the physiological status and nodulation of alfalfa plants under nutrient-poor soil stress. These positive effects could be due to the presence of protease, pectinase, chitinase, and cellulose activities together. As mentioned above, cellulase activity is involved in the nodulation process [62,63] because it facilitates the entry of the rhizobium in the root. In a similar way, pectinase can also degrade the cell wall and could increase the nodulation when plants are inoculated with the *consortium* [63]. In relation to the improvement in the germination, protease could be involved in the mobilization of the protein reserves in seeds during germination. The presence of all of the PGP properties in the *consortium* could be the reason for the increase in all of the measured parameters. The biofilm formation and the ACC deaminase activity seem to be the main differences that provide the *consortium* with the stronger effects, improving the acquisition of nutrients and decreasing the ethylene levels in plants in response to the nutrient poverty in soils [17,61]. More studies should be performed in order to elucidate the role of each PGP property in the plant and nodulation improvement.

## 4. Materials and Methods

### 4.1. Collection and Characterization of Soil

Rhizospheric soil of *Medicago* spp. plants from the Rio Piedras estuary (Huelva, Spain) (37°16′09.1″ N–7°09′36.4″ W) was collected in May 2019. Three rhizosphere samples were picked between 10–20 cm of depth from random plants with gloves and deposited into sterile bottles to be transported to the laboratory. Samples were stored at 4 °C until their utilization.

Soil samples were analyzed chemically as described [71]. The concentration of nutrients in soils was determined by inductively coupled plasma–optical emission spectroscopy (ICP–OES) (ARLFisons3410, Thermo Scientific, Walthman, MA, USA). Conductivity was measured using a Crison-522 (Crison Instrument, S.A., Barcelona, Spain) conductivity meter and redox potential and pH with a Crison pH/mVp-506 (Crison Instrument, S.A., Barcelona, Spain) portable meter [71]. Soil texture (sand, silt, and clay percentage) was determined using the Bouyoucos hydrometer method [72].

### 4.2. Isolation of Cultivable Rhizosphere Bacteria

Rhizosphere bacteria were isolated from collected soils following the protocol described in [73]. Briefly, soil samples were mixed with sterile 0.9% (*w*/*v*) saline solution by agitation for 10 min. Then, after soil decantation, supernatant (100 µL per plate) was deposited into three Petri plates containing tryptic soy agar (TSA) (Intron Biotechnology, Gyeonggi-do, Korea) medium. Inoculated plates were incubated for 72 h at 28 °C. Different bacteria were separated in different TSA plates based on the color and morphology of colonies and on the Gram staining. The pure cultures were stored in sterile 15% (*v*/*v*) glycerol at −80 °C.

### 4.3. Bacterial Screening Based on Phosphate Solubilization

In order to reduce the number of isolated bacteria with potential as PGPR, a first screening was done to select those isolates able to make phosphorus available to plants. For that, bacteria were incubated in NBRIP (National Botanical Research Institute’s phosphate growth medium) plates [74] for 5 days at 28 °C. Phosphate solubilization was observed as a transparent halo around the bacterial colony. The diameter of the halo was measured to semi-quantify the degree of phosphate solubilization.

### 4.4. Analysis of Diversity and Identification of Isolates

The diversity of P-solubilizers was first analyzed by Box-PCR. Genomic DNA was isolated from bacteria using a G-spin™ Genomic DNA Extraction (for Bacteria) kit (Intron Biotechnology, Gyeonggi-do, Korea) following the instructions supplied by the manufacturer. Box-PCR was performed using 1 µL of DNA and *Box A1R* primer (5′-CTACGGCAAGGCGACGCTGACG-3′) using the Maxime™ PCR PreMix kit (i-Taq™) (Intron Biotechnology, Gyeonggi-do, Korea) and following the next PCR conditions: initial denaturation at 94 °C for 2 min, 30 cycles of denaturation at 94 °C for 20 s, annealing at 52 °C for 20 s, extension at 72 °C for 1 min, and final extension at 72 °C for 5 min. Electrophoresis was performed in a 1.5% (*w*/*v*) agarose gel and a voltage of 70 V for 2 h.

Representative bacteria of each different Box-PCR profile were identified by 16S rRNA gene amplification using *16F27* and *16R1488* primers [73] and the Maxime™ PCR PreMix kit (i-Taq™) (Intron Biotechnology, Gyeonggi-do, Korea) following the next PCR conditions: initial denaturation at 94 °C for 2 min, 30 cycles of denaturation at 94 °C for 20 s, annealing at 58 °C for 10 s, extension at 72 °C for 50 s, and final extension at 72 °C for 5 min. Electrophoresis was performed in a 1% (*w*/*v*) agarose gel and a voltage of 120 V for 30 min. PCR products were purified with the enzyme ExoSAP-IT (Affymetrix, Santa Clara, CA, USA), following the manufacturer instructions, and sequenced by the StabVida company (Caparica, Portugal). Then, 16S rRNA gene sequences were compared with those deposited in the EzBioCloud database [75] using the Ez-Taxon-e service (www.ezbiocloud.net/eztaxon; accessed on 7 July 2021). Finally, 16S rRNA sequences were deposited in the NCBI GenBank.

### 4.5. Bacteria Characterization

#### 4.5.1. Study of Plant Growth Promoting Properties In Vitro

In addition to phosphate solubilization, indole-3-acetic acid (IAA) and siderophores production, nitrogen fixation, biofilm formation, and ACC deaminase activity were tested. IAA production was evaluated by incubating bacteria in tryptic soy broth (TSB) (Intron Biotechnology, Gyeonggi-do, Korea) supplemented with 100 mg·L^−^^1^ of L-tryptophan at 28 °C for 3 days at 200 rpm. Then, cultures were centrifuged and Salkowski reagent [76] was added to the supernatant. The change of coloration to pink was measured at 535 nm using a spectrophotometer (Lambda25; PerkinElmer, Walthmam, MA, USA), and IAA production was calculated using a pattern curve of known concentrations of IAA.

Siderophores production was tested in a Petri plate with CAS (chrome azurol S) agar medium [77] for 5 days at 28 °C in darkness. An orange halo around the bacterial growth indicated that the bacteria produced siderophores. To semi-quantify the production of siderophores, the diameter of the halo was measured.

Nitrogen fixation was determined in NFB (nitrogen free broth) plates [78]. Massive bacterial growth after 5 days at 28 °C indicated that the bacteria could fix atmospheric nitrogen.

Biofilm formation was determined following the protocol indicated by [79]. Briefly, a 96-well-plate containing TSB was inoculated with isolated bacteria and incubated at 28 °C for 4 days. After that, wells were stained with 0.01% (*w*/*v*) violet crystal for 20 min to stain the biofilm, and then biofilm was resuspended in 33% (*v*/*v*) acetic acid (in 95% (*v*/*v*) ethanol) and absorbance at 570 nm with a microplate reader ASYS UVM-340 (Montreal Biotech, Dorval, QC, Canada).

Finally, ACC deaminase activity was checked as described [80]. Succinctly, bacteria were incubated in DF (Dworkin and Foster) medium supplemented with 3 mM ACC for 24 h at 28 °C, after a previous enrichment in DF medium supplemented with (NH_4_)_2_SO_4_. Then, ACC deaminase activity was measured monitoring the α-ketobutyric acid at 540 nm in a spectrophotometer (Lambda25; PerkinElmer, Walthmam, MA, USA), and the amount of α-ketobutyric acid was calculated using a pattern curve with known concentrations. After that, the total protein concentration of bacteria was determined by the Bradford method [81], using a bovine serum albumin (BSA) standard curve. At the end, the activity of the ACC deaminase was calculated and expressed in μmoles of α-ketobutyrate per mg of protein per hour.

#### 4.5.2. Study of Enzymatic Activities

The presence of amylase, protease, chitinase, cellulase, pectinase, lipase, and DNAse were studied in the isolates. All of the activities were performed in Petri plates and incubated at 28 °C for 5 days.

Amylase activity was assayed in starch agar (Scharlab, Barcelona, Spain) plates. The plates were revealed adding lugol, and a transparent halo appeared around the bacterial biomass when bacteria had this activity. The hydrolysis of casein and lipids by proteases and lipases, respectively, were determined following the protocol described in [82]. Bacteria were incubated in casein or Tween 80 agars plates, and a halo around bacterial biomass indicated that the test was positive in both assays. Chitinase was detected with the appearance of a transparent halo around the bacterial biomass in M9 (minimal medium) plates supplemented with colloidal chitin [83]. Cellulase and pectinase activities were performed according to [84]. For cellulase, bacteria were incubated in carboxymethyl cellulose (CMC) plates and revealed with 0.1% (*w*/*v*) Congo Red and 1M NaCl. A halo around the bacterial biomass indicated that the test was positive. For pectinase activity, ammonium mineral agar (AMA) plates were used. The plates were revealed with 2% (*w*/*v*) CTAB, and positive bacteria showed a halo around.

Finally, the activity of DNAse enzyme was studied using DNA agar plates (Scharlab, Barcelona, Spain). The appearance of a transparent halo after the addition of 1 M HCl solution indicates that the bacteria are able to hydrolyze DNA.

### 4.6. Conditions and Inoculum Preparation

Three bacteria were selected based on the PGP properties and enzymatic activities to inoculate *Medicago sativa* seeds and plants and study the effect over the germination and plant development. Seeds were exposed to five conditions: C- (seeds without bacterial inoculation), L1 (seeds inoculated with *Pseudomonas* sp. L1), L2 (seeds inoculated with *Chryseobacterium soli* L2), L3 (seeds inoculated with *Priestia megaterium* L3), and CSL (seeds inoculated with a *consortium* of strains L1, L2, and L3).

On the other hand, plants were subjected to six conditions: C- (plants without bacterial inoculation), MA11 (plants inoculated with *E. medicae* MA11), L1+MA11 (plants inoculated with strains L1 and MA11), L2+MA11 (plants inoculated with strains L2 and MA11), L3+MA11 (plants inoculated with strains L3 and MA11), and CSL (plants inoculated with a *consortium* of strains L1, L2, L3, and MA11).

The inoculums for each condition were prepared by growing selected strains separately in TSB for 24 h at 28 °C and 200 rpm. Then, cultures containing 10^8^ cells·mL^−1^ were centrifuged at 8000 rpm for 10 min, washed two times with sterile 0.9 % (*w*/*v*) saline solution, and pellets were resuspended in sterile BNM (buffered nodulation medium) [85] or sterile water (depending on the experiment). For the *consortium*, the four bacteria were mixed after the individual pellet resuspension.

Culture conditions for *E. medicae* MA11 were described in [19].

### 4.7. Seed Germination Assay

To study the germination, the surface of seeds of *M. sativa* were disinfected with 70% (*v*/*v*) ethanol as described [86]. Disinfected seeds were bacterized with the corresponding inoculum by immersion into the culture for 1 h under shaking. Control seeds (C-) were immersed into sterile 0.9% (*w*/*v*) saline solution. After that, 50 bacterized seeds were plated in 0.9% (*w*/*v*) agar plates (10 seeds per plate, 5 replicates per treatment) and incubated at room temperature in darkness for 7 days. Plates were checked every day in order to determinate the kinetic of germination.

### 4.8. In Vitro Plant Cultivation

Seedlings of *M. sativa* were transferred to square plates (12 × 12 cm) containing the N-free medium BNM [85] supplemented with 0.9% (*w*/*v*) agar (10 seedlings per plate and 5 plates per treatment) and inoculated with 100 µL of the corresponding inoculum. Plates were incubated for 30 days in a plant growth chamber with 16 h light (120–130 μE m^−2^ s^−1^) at 22 °C and 8 h dark at 18 °C. After this time, the dry weight of plants and the number of nodules were recorded.

### 4.9. Greenhouse Experiments

Seedlings of *M. sativa* were transferred to plastic pots containing sterile Rio Piedras estuary sediments (2 seeds per pot and 8 pots per treatment), and pots for the same treatment were placed in a tray. The soil was sterilized three times by autoclaving 121 °C and 1 atm of overpressure for 20 min.

Plants were maintained for 60 days under greenhouse conditions and irrigated and inoculated with sterile water or the corresponding inoculum (1.25 × 10^5^ cells·g^−^^1^) every week. At the end of the experiment, the size and dry weight of plants, the number and size of leaves, and the number of nodules were recorded. Moreover, the content of nitrogen in plants was also determined using an InfrAlyzer 300 (Technicon, Tarrytown, NY, USA), as described in [87].

#### 4.9.1. Determination of Photosynthetic Parameters

Gas exchange was measured in random leaves of plants of *M. sativa* after greenhouse experiments using an infrared gas analyzer (IRGA) LI-6400 (LI-COR Biosciences, Lincoln, NE, USA) equipped with a light leaf chamber Li-6400-02B. Measurements were performed between 10 am to 2 pm hours under a photosynthetic photon flux density of 1500 µmol·m^−^^2^·s^−^^1^, a deficit of vapor pressure of 2–3 kPa, a temperature around 25 °C, and a CO_2_ concentration environment of 400 µmol·mol^−^^1^ air. The stabilization of the exchange of gases was equilibrated (120 s), and measurements were recorded to determinate the net photosynthetic rate (A_N_), stomatal conductance (g_s_), and intercellular CO_2_ concentration (C_i_).

In addition, a fluorometric analysis was performed to study the efficiency of the energy use of photosystem II (PSII). The maximum quantum efficiency of PSII photochemistry (Fv/Fm) and the quantum efficiency of PSII (Φ_PSII_) were determined using a saturation pulse method [88]. According to [89], selected leaves were dark- and light-adapted for 30 min, and then a saturating actinic light pulse of 10,000 μmol m^−2^ s^−1^ was given for 0.8 s at midday (1700 μmol photons m^−2^ s^−1^) in them using a modulate fluorimeter FMS-2 (Hansatech Instruments Ltd., Pentney, UK). With the recorded data, the electron transport rate (ETR) was calculated as described [90].

#### 4.9.2. Total Chlorophyll Content

Total chlorophyll content was extracted from 50 mg of random leaves using a mortar containing 100% acetone: 0.9% saline solution (4:1; *v*/*v*) [91]. The absorbance of the mixture was measured at 652 nm in a spectrophotometer for duplicate and the total content of chlorophyll was determined by the formula described in [92].

#### 4.9.3. Antioxidant Enzymes Measurement

Enzymatic activity of catalase (CAT), ascorbate peroxidase (APX), superoxide dismutase (SOD), and guaiacol peroxidase (GPX) were measured in the leaves of each treatment for triplicate following the methodology described by [93]. For that, random leaves from different plants of the same treatment were collected in liquid nitrogen at the end of the greenhouse experiment and stored at −80 °C until their homogenization. A total of 500 mg of vegetal material were homogenized in extraction buffer (50 mM sodium phosphate buffer; pH 7.6) and centrifuged at 4 °C and 9000 rpm for 20 min, and the vegetal extract was stored at −80 °C until their use. Total protein concentration was measured in the vegetal extract following the Bradford method [81] and using a pattern curve of BSA. According to [93], catalase activity was determined by measuring the H_2_O_2_ disappearance at 240 nm. For the ascorbate peroxidase, the oxidation of L-ascorbate was monitored at 290 nm. Superoxide dismutase was assayed using pyrogallol autooxidation at 325 nm. Finally, for the guaiacol peroxidase activity, the oxidation of guaiacol was measured at 470 nm. To determine the autooxidation of the substrates, control assays were carried out in the absence of enzymatic extract samples. The enzymatic activities were expressed as units per μg of protein.

### 4.10. Statistical Analyses

Statistical analyses were performed using the Statistica software version 6.0 (Statsoft Inc., Tulsa, OK, USA). The normality of the results was determined by the Kolmogorov-Smirnov test. Results from different treatments were compared using one-way ANOVA, and the Fisher test was carried out to determinate the statistic differences.

## 5. Conclusions

Rhizosphere of *Medicago* spp. plants in the Piedras river estuary (southwest Spain) contains PGPR with appropriate properties to be used as biofertilizers. Particularly, the strains *Pseudomonas* sp. L1, *Chryseobacterium soli* L2, and *Priestia megaterium* L3 were able to improve *M. sativa* development and nodulation in a nutrient-poor soil, acting as NER. Although single bacterial inoculation had a positive effect in *M. sativa* growth under nutrient poverty stress, the combination of the three PGPR showed the best performance, demonstrating that a *consortium* of bacteria with complementary traits works better than single inoculation. The next step to confirm the positive obtained results with the inoculation with these bacteria should be to perform a trial in the original soil without sterilization in order to observe the results in a more realistic situation. As a final conclusion, legumes inoculated with *Pseudomonas* sp. L1, *Chryseobacterium soli* L2, and *Priestia megaterium* L3 could be used as biological tools for the ecological restoration of degraded soils and to promote sustainable agriculture.

## Figures and Tables

**Figure 1 plants-11-01164-f001:**
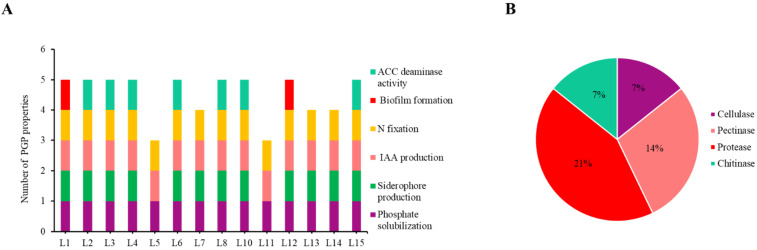
**Characterization of isolated bacteria.** (**A**) Number of PGP properties in each strain; X-axis indicated the different isolated strains. (**B**) Percentage of enzymatic activities in the isolates.

**Figure 2 plants-11-01164-f002:**
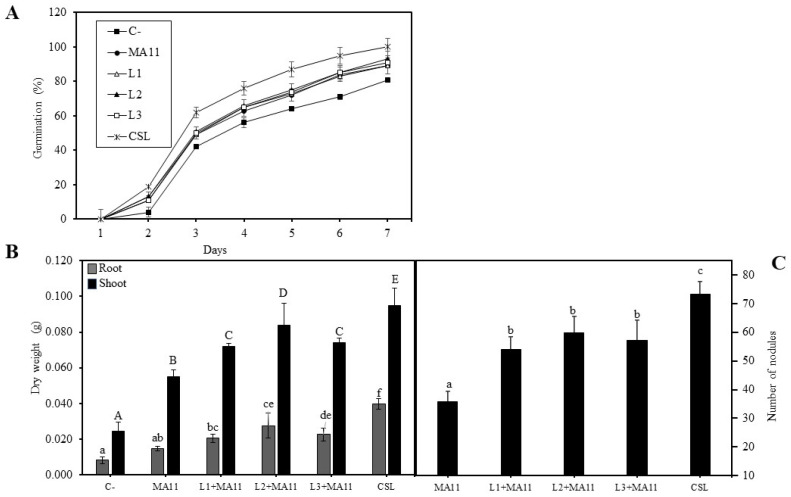
***In vitro* effects of selected PGPR in *M. sativa*.** (**A**) Number of germinated seeds in 7 days with different inoculums. Values are mean ± S.D. (*n* = 50). (**B**) Dry weight of shoots and roots of *M. sativa* plants after 30 days in BNM plates. Values are mean ± S.D. (*n* = 5). Different letters indicate means that are significantly different from each other. Lowercase and uppercase letters are used to qualify different variables and are not comparable among them (one-way ANOVA; LSD test, *p* < 0.001). (**C**) Number of nodules in plants of *M. sativa* after 30 days in BNM plates with different inoculums. Values are mean ± S.D. (*n* = 5). Different letters indicate means that are significantly different from each other (one-way ANOVA; LSD test, *p* < 0.001). C-: without inoculation; MA11: inoculation with *E. medicae* MA11; L1: inoculation with strain *Pseudomonas* sp. L1; L2: inoculation with *Chryseobacterium soli* L2; L3: inoculation with *Priestia megaterium* L3; L1+MA11: inoculation with strains L1 and MA11; L2+MA1: inoculation with strains L2 and MA11; L3+MA11: inoculation with strains L3 and MA11; CSL: inoculation with a *consortium* of strains L1, L2, L3, and MA11.

**Figure 3 plants-11-01164-f003:**
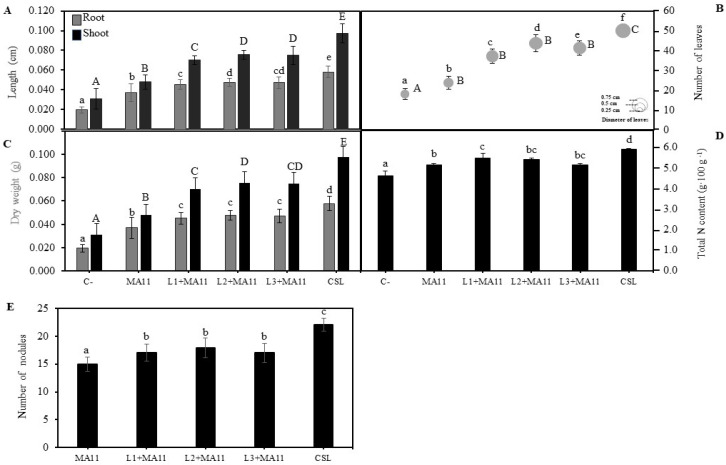
**Effects of selected PGPR in *M. sativa* under greenhouse conditions in nutrient-poor soils.** (**A**) Length of shoots and roots of *M. sativa* plants after 60 days. Values are mean ± S.D. (*n* = 16). Different letters indicate means that are significantly different from each other. Lowercase and uppercase letters are used to qualify different variables and are not comparable among them (one-way ANOVA; LSD test, *p* < 0.001). (**B**) Number and size of leaves in plants of *M. sativa* after 60 days from germination. Values are mean ± S.D. (*n* = 16). The size of the circles indicates the mean of the diameter of the leaves. Different letters indicate means that are significantly different from each other. Lowercase letters are used to qualify the number of leaves per plant and uppercase letters are used to qualify the diameter of leaves and are not comparable among them (one-way ANOVA; LSD test, *p* < 0.0001). (**C**) Dry weight of shoots and roots of *M. sativa* plants after 60 days from germination. Values are mean ± S.D. (*n* = 16). Different letters indicate means that are significantly different from each other. Lowercase and uppercase letters are used to qualify different variables and are not comparable among them (one-way ANOVA; LSD test, *p* < 0.001). (**D**) Nitrogen content in plants of *M. sativa* after 60 days from germination. Values are mean ± S.D. (*n* = 16). Different letters indicate means that are significantly different from each other. Lowercase and uppercase letters are used to qualify different variables and are not comparable among them (one-way ANOVA; LSD test, *p* < 0.001). (**E**) Number of nodules in plants of *M. sativa* after 60 days from germination. Values are mean ± S.D. (*n* = 16). Different letters indicate means that are significantly different from each other (one-way ANOVA; LSD test, *p* < 0.0001). C-: without inoculation; MA11: inoculation with strain MA11; L1+MA11: inoculation with strains L1 and MA11; L2+MA1: inoculation with strains L2 and MA11; L3+MA11: inoculation with strains L3 and MA11; CSL: inoculation with a *consortium* of strains L1, L2, L3, and MA11.

**Figure 4 plants-11-01164-f004:**
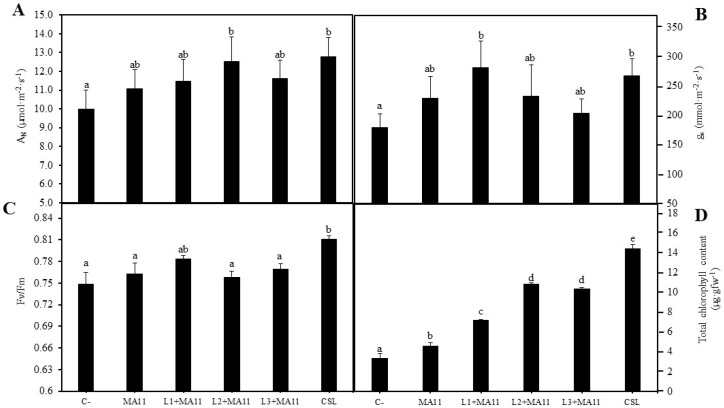
**Photosynthetic parameters.** (**A**) Net photosynthetic rate, (**B**) stomatal conductance, (**C**) maximum quantum efficiency of PSII photochemistry, and (**D**) total chlorophyll content in plants of *M. sativa* after 60 days from germination under greenhouse conditions with a nutrient-poor soil as substrate and different inoculation treatments. Values are mean ± S.D. (*n* = 16). Different letters indicate means that are significantly different from each other (one-way ANOVA; LSD test, *p* < 0.005). C-: without inoculation; MA11: inoculation with strain MA11; L1+MA11: inoculation with strains L1 and MA11; L2+MA1: inoculation with strains L2 and MA11; L3+MA11: inoculation with strains L3 and MA11; CSL: inoculation with a *consortium* of strains L1, L2, L3, and MA11.

**Figure 5 plants-11-01164-f005:**
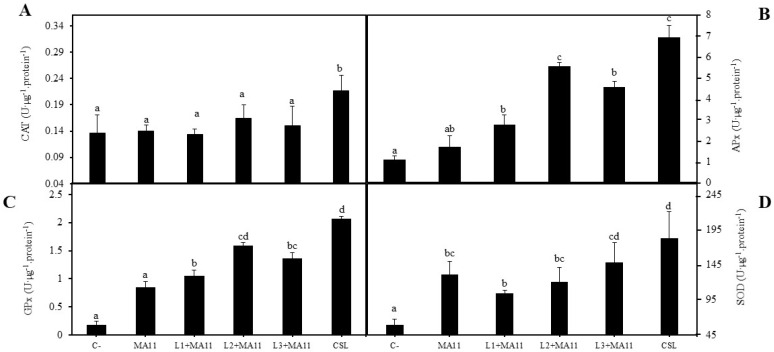
**Level of antioxidant activities.** (**A**) Catalase, (**B**) ascorbate peroxidase, (**C**) guaiacol peroxidase, (**D**) superoxide dismutase activities in leaves of *M. sativa* 60 days after germination under greenhouse conditions with a nutrient-poor soil as substrate and different inoculation treatments. Values are mean ± S.D. (*n* = 16). Different letters indicate means that are significantly different from each other (one-way ANOVA; LSD test, *p* < 0.001). C-: without inoculation; MA11: inoculation with strain MA11; L1+MA11: inoculation with strains L1 and MA11; L2+MA1: inoculation with strains L2 and MA11; L3+MA11: inoculation with strains L3 and MA11; CSL: inoculation with a *consortium* of strains L1, L2, L3, and MA11.

**Table 1 plants-11-01164-t001:** Physicochemical properties and nutrient concentrations of soil from the Rio Piedras estuary.

Physicochemical Properties					
Texture (%) *	Organic Material (%)			Conductivity (mS·cm^−1^)	pH
92/4/4	0.98 ± 0.01			1.245 ± 0.009	8.5 ± 0.014
**Nutrient concentration (mg·kg^−1^)**					
**Ca**	**K**	**Mg**	**Mn**	**Na**	**P**
3.162 ± 0.448	0.253 ± 0.021	0.277 ± 0.029	189.822 ± 2.860	0.159 ± 0.006	0.014 ± 0.002

Values are mean ±S.D. (*n* = 3). * Texture (sand/slit/clay percentages).

**Table 2 plants-11-01164-t002:** Identification of cultivable bacteria isolated from the rhizosphere of *Medicago* spp. Using the EzBiocloud database. Selected strains appear in bold.

Strain	Sequence Size (bp)	Related Species	% ID	Accession Number
**L1**	**469**	** *Pseudomonas kitaguniensis* **	**98.01**	**OM397092**
**L2**	**1272**	** *Chryseobacterium soli* **	**99.76**	**OM397093**
**L3**	**1366**	** *Priestia megaterium* **	**100**	**OM397094**
L6	1298	*Pseudomonas canadensis*	100	OM397096
L7	1270	*Bacillus paramycoides*	100	OM397095
L10	1293	*Pseudomonas baetica*	99.77	OM397097
L12	1277	*Pseudomonas moorei*	99.76	OM397098
L13	1301	*Pseudomonas moorei*	99.69	OM397099
L14	937	*Pseudomonas moraviensis*	99.89	OM442986
L15	1317	*Buttiauxella noackiae*	99.92	OM397100

## Data Availability

Not applicable.

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
