# Peer review of "Role of Nodulation-Enhancing Rhizobacteria in the Promotion of *Medicago sativa* Development in Nutrient-Poor Soils"

_plants, 2022, doi:10.3390/plants11091164_

Round 1
Reviewer 1 Report
The subject of the work is the study and characterization of nodulation-enhancing rhizobacteria for the improvement of the growth of the Medicago sativa species in degraded and, therefore, nutrient-poor soils. The problem of marginal soils is constantly increasing, and the severe global water crisis has a significant impact on the degradation of agricultural soils. The topic discussed is, therefore, substantial and interesting.
General comments: the work is well structured, well detailed with good figures and tables. Here are some specific comments.
Keywords:
PGPR and NER are entered for the first time. Perhaps it is better to specify the acronym here.
Materials and Methods:
Line 465-470: "Then, cultures containing 108 cells·ml-1 466 were centrifuged at 8000 rpm for 10 minutes, washed 2 times with sterile 0.9 % (w/v) saline solution, and pellets were resuspended…..". I did not find how many CFU / g of soil was then used.
Line 490-491: "Soil was sterilized three times by autoclaving 121 ˚C and 1 atm 490 of overpressure for 20 minutes". Allow me to raise some doubts about the soil sterilization under investigation. I understand that this can better highlight the effect of the inocula. But this is an unrealistic situation. In an actual condition, in the soil, the interactions between the rhizosphere microorganisms can be fundamental.
Also, for this reason, in the comment to the conclusions, I suggest providing a field test.
Results:
Figure 1 (Box-PCR dendrogram) could be moved to the Supplementary Materials.
Line 291-293: "This property was observed in all isolates making all them good candidates to improve the nodulation of legumes growing in degraded estuaries". I agree with this statement. Thus I wonder why to choose only the three strains showing proteolytic activities. Action against phytopathogens in abiotic stress conditions is critical, but the primary purpose of the research is the selection of Nodulation-Enhancing Rhizobacteria (NER), and other strains among the 14 isolates have been shown to possess beneficial plant growth-promoting properties.
Conclusions:
I always appreciate short conclusions, especially if the discussion is detailed enough. But I think it is important to add what you intend to do after. What is the way forward you have defined? In my opinion, this should be a field test, necessary to understand if the selected inocula can have the same positive effect obtained in the greenhouse.

Reviewer 2 Report
Dear Editor
I write in reference to the paper plants-1664739 “Role of nodulation-enhancing rhizobacteria in the promotion of Medicago sativa development in poor-nutrient soils”.
The paper describes the characterization of the plant growth promoting properties of 14 bacterial strains isolated from the rhizosphere of Medicago sativa plants grown in degraded soils, and the performance of three of them as promoters of M. sativa growth and nodulation.
Although the topic is not completely novel, the effect of nodulation-enhancing rhizobacteria in legumes growth on poor soils is not well explored. In my opinion, the manuscript is well written, most methods are appropriated and the conclusions are supported by the results.
I strongly suggest the elimination of the Box-PCR experiment plus some minor changes before the publication of the manuscrit.
Specifically:
Abstract
Lines 21-22. “...activating the antioxidant system before non-inoculated plants”. The authors did not perform a kinetic assay. Please correct.
Introduction
Line 37. Delete a dot (etc..).
Results
Lines 111-119. The problem with Box-PCR experiment is that it was employed for group strains in bases to genetic similarity even to the level of species and strains, but further experiment (Table 2) showed that it did not work. The branches of figure 1 did not reproduce the known taxonomy and do not show to be a good criteria to grope bacteria strains.
Why did Pseudomonas not grouped together? Why L13 and L14 ( Pseudomonas, Proteobacteria) are grupped with L3 (Priestia Firmicute) and not with L6, L10 and L12? Why L2 (Pseudomonadaceae) is grupped with Butiauxiella (Enterobacteriaceae) and not with the other Pseudomonadace? Why L2 (Chryseobacterium, Bacteroidetes) is grouped with L1 (Pseudomonas, Proteobacteria)?
In axis X of figure 1, the genetic distance does not concord with the known taxonomy.
Maybe L6, L7 and L8 are the same species, in absence of controls to validate the method the figure 1 only suggests and in general is in contradiction with table 1.
I Strongly suggest the elimination of the Box-PCR experiment and the adequation of the text to this elimination.
Line 117-118. It must say Table 2.
Line 143. Please change to “that showed the tested enzymatic activities”.
Line 149. Is the consortium the CSL treatment? If it is the case please define the term anb be consistent throughout the text.
Line 166. Figure 3 legend and line 208 (figure 4 legend). Please mention that lowercase and uppercase letters are used to qualify different variables and are not comparable among them.
Line 183. Please define CSL. Is it the consortium? Please explain.
line 188. Please review the statistical treatment in panel S2A. The letter "b" is not necessary.
Discussion
Line 235. I do not understand the argument. How are related ecological values of estuaries with loss of diversity or low fertility?
Line 381. If the Box-PCR would not be eliminated. Please provide the method and primer references.
Reviewer 3 Report
The evaluation of the article :”Role of nodulation-enhancing rhizobacteria in the promotion of Medicago sativa development in poor-nutrient soils” Special Issue: “Plant–Microbe Interactions for Sustainable Agriculture”:
General Aspects:
- The general content of the article is adequate and complete.
- The number of authors is adequate (in relation to the work carried out).
- The language is correct. The wording is adequate and easy to understand.
- The structure of the text is correct.
- The bibliography is up-to-date and adequate.
- Suggestions:
- The summary does not clearly state the hypothesis or the objective of the trial.
- A clearer and more concrete wording of the conclusions is suggested.
- A more extensive review of recent texts (2022) is advisable.
Reviewer 4 Report
In the present manuscript the authors isolated and selected several bacterial strains to investigate their effect on the nodulation and plant growth paramteres of Medicago sativa.
The experiments were well-done and the results are interesting, but the presentation of the results, their description and interpretation need improvement.
Several aspects mentioned in the discussion should be mentioned in the introduction or results part - the authors describe the role of IAA, siderophores, ACC deaminase etc. in the discussion. However, this basic knowledge is needed to understand the motivation of the authors why they did these measurements and should not only be mentioned at the end of the manuscript.
In the abstract please specify the number of growth promoting factors tested (e.g. the strains showed at least 3 out of x plant growth promoting properties tested).
Please give sufficient details in the figure legends so that figures can be understood without reading the whole text.
Figure 1 - Please mention what L14, L3 etc. mean and where/how they were isolated.
Please delete the boxes around all figures and figure legends.
Why do you use the term aerial part - from my botanical understanding a plant consists of roots and shoots.
Figure 3: Please move A to the left.
Figure 4 Please mention what C-, MA11 etc mean in the legend. I do not understand the double letters (capital letters and small letters) in B.
Please use: 60 days after inoculation or germination
When the same x-axis is used and samples are the same please only show it once (e.g for length and dry weight and Total N content)
Please move 4E to the left.
Overall, the text contains many expressions, half sentences etc. that are not correct. I assume this is mainly due to difficulties of writing in English. I list below only very few examples but the whole manuscript needs substantial polishing by a native speaker, and cannot be published without these revisions.
Please correct throughout the text - poor-nutrient to nutrient-poor.
line 18 delete and
line 19 numbers
line 22 used compared to instead of before, delete and
line 23 plant
line 32 the genera
line 34 plants offer ...and the bacteria provide plants reduced nitrogen due to the ability of the rhizobia to fix....
line 44 don't use the word foodstuff in a scientific paper
....
Round 2
Reviewer 4 Report
The authors partially addressed the reviewers comments. Figure legends provide more details and the authors did some corrections on the text but they did not ask a native speaker to polish the manuscript. Hence, the manuscript still needs substantial polishing not only with regard to English but also in terms of scientific descriptions. It is scientifically not sound to speak of e.g. best results, best values, best behaviour etc. How do you judge if a result is good or bad? Values can be higher or lower, plants bigger or smaller but not better. Furthermore, what do you mean by enhancement in plant physiology and improved physiological parameters. These are two striking examples but more sentences require better scientific writing which makes the whole text hard to read and the authors should seek for help.
In Figure 3 you point out that you did two independent statistical analysis, but you have to mention which values indicate which comparison.
Please reduce the use of "besides" - the reader knows what you wrote in the previous sentence or paragraph and it does not have to be repeated.
In the discussion the authors mention that Pseudomonas was the most abundant bacterial genus they isolated. It is fully correct that it is ubiquitous but its high abundance when using culturable approaches to characterize the microbial communities could also be due to a methodological bias - other bacteria are not culturable and hence missing or they need more time to establish. Scientifically, I also miss a better discussion of the use of single bacterial inoculants vs. consortia. Why do you see stronger effects after inoculation with a consortium? What could be the mechanism?
Round 3
Reviewer 4 Report
Dear authors - I appreciate that you made another effort to improve the manuscript. Unfortunately, I do not know whom you have contacted to revise and improve your manuscript. Some paragraphs have definitely improved, but for some sentences it would have been better to keep the previous version. Overall, the manuscript still contains a large number of small mistakes. Additionally, though you have now corrected the few sentences that I pointed out additional sentences lack the standards for publication in a scientific journal. An example is lines 271-285. I can guess what you want to describe but it does not read like a scientific description of results. There are many more. As mentioned in my previous reviews I strongly recommend to get a native speaker familiar with proof-reading to go through your manuscript.
Furthermore, you have added now a new paragraph on the advantages of using a consortium instead of a single strain inoculation. I strongly miss a comparison with previous studies using consortia vs. single isolates for inoculation.